# The Unfolding Space Glove: A Wearable Spatio-Visual to Haptic Sensory Substitution Device for Blind People

**DOI:** 10.3390/s22051859

**Published:** 2022-02-26

**Authors:** Jakob Kilian, Alexander Neugebauer, Lasse Scherffig, Siegfried Wahl

**Affiliations:** 1Köln International School of Design, TH Köln, 50678 Köln, Germany; jkilian@kisd.de (J.K.); lscherff@kisd.de (L.S.); 2ZEISS Vision Science Laboratory, Eberhard-Karls-University Tübingen, 72076 Tübingen, Germany; a.neugebauer@uni-tuebingen.de; 3Carl Zeiss Vision International GmbH, 73430 Aalen, Germany

**Keywords:** tactile vision sensory substitution, blind, visually impaired, mobility, navigation, orientation, locomotion, open source, haptic, wearable

## Abstract

This paper documents the design, implementation and evaluation of the Unfolding Space Glove—an open source sensory substitution device. It transmits the relative position and distance of nearby objects as vibratory stimuli to the back of the hand and thus enables blind people to haptically explore the depth of their surrounding space, assisting with navigation tasks such as object recognition and wayfinding. The prototype requires no external hardware, is highly portable, operates in all lighting conditions, and provides continuous and immediate feedback—all while being visually unobtrusive. Both blind (n = 8) and blindfolded sighted participants (n = 6) completed structured training and obstacle courses with both the prototype and a white long cane to allow performance comparisons to be drawn between them. The subjects quickly learned how to use the glove and successfully completed all of the trials, though still being slower with it than with the cane. Qualitative interviews revealed a high level of usability and user experience. Overall, the results indicate the general processability of spatial information through sensory substitution using haptic, vibrotactile interfaces. Further research would be required to evaluate the prototype’s capabilities after extensive training and to derive a fully functional navigation aid from its features.

## 1. Introduction

### 1.1. Navigation Challenges for Blind People

Vision is the modality of human sensory perception with the highest information capacity [1,2]. Being born blind or losing one’s sight later in life involves great challenges. The ability to cope with everyday life independently, to be mobile in unfamiliar places, to absorb information and, as a result, to participate equally in social, public and economic life can be severely hampered, ultimately affecting one’s quality of life [3,4,5,6]. Society can, of course, address this issue on many levels, for example by ensuring accessibility of information or by designing public spaces to meet the specific needs of blind individuals or, more generally, visually impaired people (VIPs). In addition, technical aids, devices and apps are constantly being developed to assist VIPs with certain tasks [7,8]. These aids can essentially be divided into three aspects: obtaining information of the surroundings (What is written here? What is the object ahead like?), interfacing with machines and computers (input and output) and navigation (How do I get there? Can I walk this way?). While there are certainly large overlaps between these three aspects, this paper exclusively focuses on the third—navigation.

Following the definition of Montello and Sas [9], navigation (often used synonymously with mobility, orientation or wayfinding [10,11]) is the ability to perform “coordinated and goal-directed movement[s] through the environment”. It can be divided into two sub-components: *wayfinding* (long-range navigation to a target destination, spatial orientation, knowledge about surroundings and landmarks outside the immediate environment) and *locomotion* (short-range navigation, moving in the intended direction without colliding or getting stuck, obstacle detection and development of strategies to overcome them). This paper will concentrate on the latter and discuss how VIPs can be supported in this by technical aids—that is, without the help of human companions.

### 1.2. Ways to Assist Locomotion

One theoretical solution would be to rehabilitate (rudimentary) vision: since the 1990s, research has been conducted into surgical measures in both retinal implants stimulating the optic nerve and brain implants attached directly to the visual cortex. The quality of vision restored by this kind of surgery varies widely and, even in the best cases, represents only a fraction of the visual acuity of people with ordinary eyesight. Together with high costs, this leads to the fact that invasive measures of this kind are still far from widespread use and have so far only been tested on a small number of people. In the medium term, however, improving the quality of implants and simplifying surgical procedures could make the technology available to a wider public [12,13]. With regard to navigation, however, it is questionable whether the mere provision of visual information to the brain would adequately address the problem. Even if sighted individuals can rely on the visual modality for the most part, navigation and the acquisition of spatial knowledge required for it are by no means dedicated visual tasks. Blind people (but also sighted people to some extent) use multiple modalities and develop various strategies to master navigation and locomotion, which can be described as multimodal or even amodal task-specific functions. There is an ongoing discussion about how exactly VIPs—a very heterogeneous group with varying capacities for absorbing environmental information—obtain and cognitively process spatial knowledge [14,15].

Two well established and commercially available aids that do not attempt to assist locomotion through vision restoration alone are the white long cane (which provides basic spatial information about objects in close proximity) and the guide dog (which uses the dog’s spatial knowledge to assist navigation).

Figures for prevalence of the *white cane* among VIPs vary greatly depending on the age of the study population, the severity of their visual impairment and other factors [16,17,18,19], but in one example (USA, 2008) it is as low as ~10% [16]. Even though the white cane, once mastered, has proven to be an extremely helpful (and probably the most popular) tool for VIPs, it comes with the drawback of having only a limited range (approx. 1 m radius) and not being able to recognise aerial obstacles (tree branches, protruding objects) [20]. Smart canes that give vibratory feedback, capable of recognising objects above waist height or further away, could offer a remedy, but have, to the best of our knowledge, not yet achieved widespread use. The reasons for this, as well as their advantages and disadvantages, have been discussed in various publications [20,21,22,23,24].

*Guide dogs*, on the other hand, are another promising option that can bring further advantages for blind people besides solving navigation tasks [25,26,27,28]. However, they are even less widespread (e.g., only 1% of the blind people in USA, 2008 [16] or 2.4% of “the registered blind” in the U.K. [26] owned one). The reasons for this and the drawbacks of the guide dog as a mobility aid are manifold and have been frequently discussed in the literature.

Even among the users of these two aids, 40% reported head injuries at least once a year (18% once a month) and as a result, 34% leave their usual routes only once or several times a month, while 6% never do [19].

With an estimated worldwide total number of 295 million people with moderate or severe visual impairment in 2020, of which 36 million are totally blind [29], it can be assumed that the locomotion navigation tasks addressed—among others—still represent a global problem for VIPs and pose an important field of research.

### 1.3. Sensory Substitution as an Approach

The concept of Sensory Substitution (SS) offers a promising approach to address this shortcoming. The basic assumption of SS is that the function of a missing or impaired sensory modality can be replaced by stimulating another sensory modality using the missing information. This only works because the brain is so plastic that it learns to associate the new stimuli with the missing modality, as long as they fundamentally share the same characteristics [30]. Surgical intervention would not be necessary because existing modalities or sensory organs can be used instead.

There is a great amount of scientific work on the topic of SS, specifically dealing with the substitution of visual information. The research field was established in the 1960s by a research group around Paul Bach-y-Rita; they developed multiple variations of a Sensory Substitution Device (SSD) stimulating the sense of touch in order to replace missing vision, commonly called Tactile-Vision Substitution Systems (TVSS) [31,32,33]. Many influential publications followed this example and developed SSDs for the tactile modality [34,35,36,37], one even suggesting using a glove as tactile interface [38]; others addressed further modalities, such as the auditory with so-called Auditory-Vision Substitution Systems (AVSS) [39,40,41,42,43]. While there are summaries of existing approaches [44,45], there are also many other, smaller publications on the topic in the literature—often only looking at a sub-area of SS.

It should be noted that there is further work on so-called Electronic Travel Aids (ETAs) [24,44,46]. SSDs differ from them in the sense that they pass largely unprocessed information from the environment on to the substituting sensory modality and leave the interpretation to the user, while ETAs provide pre-interpreted abstract information (e.g., when to turn right or where a certain object is located).

Additionally, there is also research on SS-like assistive devices outside the scientific context [47,48,49,50].

### 1.4. Brain Plasticity and Sensory Substitution

The theoretical basis of SS is summarised under the term of brain plasticity. Although the focus of this paper is not on the neurophysiological discussion of this term, a brief digression is nevertheless helpful in order to understand some of the design decisions made in this project.

In general, *brain plasticity* describes the “adaptive capacities of the central nervous system” and “its ability to modify its own structural organization and functioning” [30]. While neuroscience has long assumed a fixed assignment of certain sensory and motor functions to specific areas of the brain, we today know that the brain is capable of reorganising itself, e.g., after brain damage [51] and, moreover, is capable of learning new sensory stimuli not only in early development but throughout life [52]. For sensory substitution to work and for the new neural correlate to be learned, a number of conditions are nevertheless necessary; Bach-Y-Rita et al. [34] point out that there has to be

“(a) functional demand, (b) the sensor technology to fill that demand, and (c) the training and psychosocial factors that support the functional demand”.

An SSD only needs a sensor that picks up the information, an interface that transmits it to human receptors, and finally and very importantly, the possibility for the user to modify the sensory stimulus by motor action in order to determine the initial origin of the information [34]. The importance of the latter close dependence between motor action and sensory perception has been emphasised in many publications [33,34] and is assumed to be the basis of *any* sensory experience [53].

There still is a vital discussion across disciplines about how cognitive processing of sensory stimuli is carried out by the brain. Worth mentioning here is the Theory of Sensorimotor Contingencies (SMCs) dismissing longstanding representational models and describing the supposedly passive perception of environmental cues as an active process that relies on regularities between action and reception that have to be learned [54]. The literature on SSDs and the SMC theory mutually refer to each other [54,55].

### 1.5. Pitfalls of Existing Substitution Systems

Despite the long tradition of research on the topic of SS and numerous publications with promising results, the concept has not yet achieved a real breakthrough. The exact reasons for the low number of available devices and users have often been discussed and made the subject of proposals for improvement [30,40,44,55,56].

Certain prerequisites that an SSD must meet in order to be used by a target group can be gathered from both existing literature and methods of interaction design. These aspects are of a very abstract nature and their implementation in practice is challenging and often only partially achievable. The following 14 aspects or prerequisites—the first ten of which were originally formulated as *problems* of existing SSDs by Chebat et al. [55]—were taken into account in the design and evaluation of the proposed SSD: learning, training, latency, dissemination, cognitive load, orientation of the sensor, spatial depth, contrast, resolution, costs, motor potential, preservation of sensory and motor habits, user experience and joy of use, and aesthetic appearance. See Appendix B for a discussion.

The motivation to deal with this field arose from the technological progress since Bach-y-Rita’s early pioneering work and his analogue experimental set-ups; improvements in price, portability and processing power of modern digital computing devices, sensors and other components necessary for the development of an SSD have opened up many new possibilities in the field and facilitate the implementation of these 14 aspects, now more than ever. Recent literature on assistive technology for VIPs also suggests that the field is “gaining increasing prominence owing to an explosion of new interest in it from disparate disciplines” having a “very relevant social impact” [57].

## 2. The Unfolding Space Glove

### 2.1. Previous Work

The Unfolding Space Glove is an SSD; it translates spatio-visual depth images into information that can be sensed haptically. In a broader sense, it is a TVSS, with the original term *tactile* perception deliberately being changed to *haptic* because of the high level of integration of the motor system, and the term *visual* being changed to *spatio-visual* to describe the input more accurately.

It was first drafted in previous work by Kilian in 2018 [58,59] with a focus on Interaction Design (only available in German). However, the first prototypes of the glove were still a bit cumbersome, heavy, had higher latencies and were prone to errors. Nevertheless, they were able to prove the functional principle and demonstrate that more research on this device is worthwhile. Through the course of this project, the device was refined and the prototype tested in this study was ultimately able to deliver promising results in the empirical tests (see results and discussion section) and meets many of the previously defined prerequisites (see discussion section). For more details and background information on the project please also see the project website https://www.unfoldingspace.org (accessed on 26 January 2022). Code, hardware, documentation and building instructions are open source and available in the public repository https://github.com/jakobkilian/unfolding-space (accessed on 26 January 2022). Consider Release v0.2.1 for the stable version used in this study and consider more recent commits in which the content has been revised for better accessibility.

### 2.2. Structure and Technical Details

The Unfolding Space Glove (Figure 1) essentially consists of two parts: a USB power bank that is worn on the upper arm (or elsewhere) and the glove itself, holding the camera, actuators, computing unit and associated technology. The only connection between them is the power supply USB cable. A list of the required components and photographic documentation of the assembly process is attached in the Appendix A and can be found in the aforementioned Github repository. The mere material costs of the entire set-up are about $600, of which about two thirds go to the camera alone (see Appendix C).

Structurally, a TVSS typically consists of three components: the input (a camera that captures images of the environment), the computing unit (translating these images into patterns suitable for tactile perception), and finally the output (a tactile interface that provides this information to the user).

The selected input system gathering 3D information on the environment is the “Pico Flexx” Time of Flight (ToF) camera by *pmdtechnologies* [60]. ToF refers to a technique that determines the distance to objects by measuring the time that actively emitted light signals take to reach them (and bounce back to the sensor). For quite a while, SS research focused on conventional two-dimensional greyscale images, and it is only in the last few years that the use of 3D images or data, especially from ToF cameras, has been investigated. Due to the advantages of today’s ToF technology compared to other 3D imaging methods (such as structured light or stereo vision) [61], it seemed reasonable to make efforts to explore exactly this combination. See Appendix C for a more detailed discussion of existing 3D SSDs and the choice of the ToF camera technology.

The computing unit, a Raspberry Pi Compute Module 4, is attached to the glove as part of the *Unfolding Space Carrier Board* which is described more closely in Appendix D.

The output is a 3 × 3 matrix of (vibrating) linear resonant actuators (LRAs) placed on the back of the hand in the glove. The choice of actuators and the reasons for positioning them on the back of the hand are described in Appendix E.

### 2.3. Algorithm

The SSD’s translation algorithm takes the total of 224 × 171 (depth) pixels from the ToF camera and calculates the final values of the glove’s 3 × 3 vibration pattern. Each motor represents the object that is closest to the glove within the corresponding part of the field of view of the camera. A detailed explanation of the algorithm can be found in Appendix F. The code files and their documentation are attached as zip files in the Appendix A but can also be accessed in the aforementioned Github repository.

### 2.4. Summary

The resulting system achieves a frame rate of 25 fps and a measured latency of about 50 ms. About 10 ms of this is due to the rise time of the LRAs, 3 ms to the image-processing on the Raspberry Pi and an unknown part to the operations of the Pico Flexx specific library. The system has a horizontal field of view of 62° and a vertical of 45° with a detection range from 0.1 m to 2 m [60]. The beginning of the detection range is determined by the limitations of the ToF method, while the 2 m maximum distance was just fixed for this study and could be adjusted in the future (maximum of the camera is 4 m [60] with decreasing quality in the far range). The glove by itself weighs 120 g with all its components, the power bank and bracelet weigh 275 g together, giving a total system weight of 395 g. The glove was produced in two sizes, each with a spare unit, in order to guarantee a good fit for all subjects and a smooth conduct of the study.

Now that the physical system has been described in detail, the next section will explain the study design.

## 3. Experimental Methods

### 3.1. Ethics

In order to evaluate the SSD described, a quasi-experiment was proposed and approved by the ethics committee of the faculty of medicine at the university hospital of the Eberhard-Karls-University Tübingen in accordance with the 2013 Helsinki Declaration. All participants were informed about the study objectives, design and associated risks and signed an informed consent form to publish pseudonymous case details. The individuals shown in photographs in this paper have explicitly consented to the publication of those by signing an additional informed consent form.

### 3.2. Hypotheses

The study included training and testing of the white long cane. This was not done with the intention of pitting the two aids against each other or eventually replacing the cane with the SSD. Rather, the aim was to be able to discuss the SSD comparatively with respect to a controlled set of navigation tasks. In fact, the glove is designed in a way that it could be worn and used in combination with the cane in the other hand. Testing not only both aids but also the combination of both would, however, introduce new unknown interactions and confounding factors. The main objective of this study thus reads: “The impact of the studied SSD on the performance of the population in both navigation and obstacle detection is comparable to that of the white long cane.” The hypothesis derived from this is complex due to one problem: blind subjects usually have at least basic experience with the white cane or have been using it on a daily basis for decades. A newly learned tool such as the SSD can therefore hardly be experimentally compared with an internalised device like the cane. A second group of naive sighted subjects was therefore included to test two separate sub-hypotheses:

**Hypothesis** **1.**
*
**Performance.**
*


**Sub-Hypothesis** **H1a.**
***Non-Inferiority of SSD Performance:** after equivalent structured training with both aids, sighted subjects (no visual impairment but blindfolded, no experience with the cane) achieve a non-inferior performance in task completion time (25 percentage points margin) with the SSD compared to the cane in navigating an obstacle course.*


**Sub-Hypothesis** **H1b.**
***Equivalency of Learning Progress across Groups:** at the same time, blind subjects who have received identical training (here only with the SSD) show equivalent learning progress with the SSD (25 percentage points margin) as the sighted group.*


With both sub-hypotheses confirmed, one can therefore, in simple terms, make assumptions about the effect of the SSD on the navigation of blind people compared to the cane, if both had been learned similarly. In addition, two secondary aspects should be investigated:

**Hypothesis** **2.**
*
**Usability & Acceptance.**
*
*The device is easy to learn, simple to use, achieves a high level of user enjoyment and satisfaction and thus strong acceptance rates.*


**Hypothesis** **3.**
*
**Distal Attribution.**
*
*Users report unconscious processing of stimuli and describe the origin of these haptic stimuli distally in space at the actual location of the observed object.*


### 3.3. Study Population

A total of 14 participants were recruited mainly through calls at the university and through local associations for blind and visually impaired people in Cologne, Germany. Appendix G contains a summary table with the subject data presented below. The complete data set is also available in the study data in Appendix A.

Six of the subjects were normally sighted and had a visual acuity of 0.3 or higher; eight were blind (congenitally and late blind), thus had a visual acuity of less than 0.05 and/or a visual field of less than 10° (category 3–5 according to ICD-10 H54.9 definition) on the better eye. Participants’ self-reports about their visual acuity were confirmed with a finger counting test (1 m distance) and, if passed, with the screen based Landolt visual acuity test “FrACT” [62] (3 m distance) using a tactile input device (Figure 2A).

Two subjects were excluded from the evaluation despite having completed the study: on average, subject f (cane = 0.75, SSD = 2.393) and subject z (cane = 1.214, SSD = 1.500) caused a remarkably higher number of contacts (two to three-fold) with *both* aids than the average of the remaining blind subjects (cane = 0.375, SSD = 0.643). For the former this can be explained by a consistently high level of nervousness when walking through the course. With both aids, the subject changed their course very erratically in the event of a contact, causing further contacts or even collisions right away. The performance of the latter worsened considerably towards the end of the study, again with both aids, so much so that the subject was no longer able to fulfil the task of avoiding obstacles at all, citing “bad form on the day” and fatigue as reasons. In order to not influence the available data by this apparent deviation, these two subjects were excluded from all further analysis.

The age of the remaining participants (six female, five male, one not specified) averaged 45 ± 16.65 years and ranged from 25 to 72 years. All were healthy—apart from visual impairments—and stated that they were able to assess and perform the physical effort of the task; none had prior experience with the Unfolding Space Glove or other visual SSDs.

All participants in the blind group have been using the white cane on a daily basis and for at least five years and/or did an Orientation and Mobility (O&M) training. Some occasionally use technical aids like GPS-based navigation and one even had prior experience using the *feelspace* belt (for navigation reasons only, not for augmentation of the Earth’s magnetic field). Two reported to use *Blind Square* from time to time and one used a monocular.

None of the sighted group had prior experience with the white cane.

### 3.4. Experimental Setup

The total duration of the study per subject differs between the blind and the sighted group, as the sighted have to do the training with both aids and the blind with the SSD only (since one inclusion criterion was experience in using the cane). The total length thus was about 4.5 h in the blind group and 5.5 h in the sighted group.

In addition to paper work, introduction and breaks, participants of the sighted group received 10 min of an introductory tutorial on both aids, had 45 min of training with them, spent 60 min using them during the trials (varied slightly due to the time required for completion) and thus reached a total wearing time of about 2 h with each aid. In the blind group, the wearing time of the SSD was identical, while the wearing time of the cane is lower due to the absence of tutorial and training sessions with it.

The study was divided into three study sessions, which took place at the Köln International School of Design (TH Köln, Cologne, Germany) over the span of six weeks. In the middle of a 130 square meter room, a 4 m wide and 7 m long obstacle course was built (Figure 2B), bordered by 1.80 m high cardboard side walls and equipped with eight cardboard obstacles (35 × 31 × 171 cm) distributed on a 50 cm grid (Figure 2C) according to the predefined course layouts.

### 3.5. Procedure

Before the first test run (baseline), the participants received a 10-min *Tutorial* in which they were introduced to the handling of the devices. Directly afterwards, they had to complete the first *Trial Session (TS)*. This was followed by total of three *Practices Sessions (PS)*, each of them being followed by another TS—making the Tutorial, four TS and three PS in total. The study concluded with a questionnaire at the end of the third study session after completion of the fourth and very last TS. An exemplary timetable of the study procedure can be found in the Appendix A.

#### 3.5.1. Tutorial

In the 10-min Tutorial, participants were introduced to the functional design of the device, its components and its basic usage such as body posture and movements while interacting with the device. At the end, the participants had the opportunity to experience one of the obstacles with the aid and to walk through a gap between two of these obstacles.

#### 3.5.2. Trial Sessions

Each TS consisted of seven consecutive runs in the aid condition cane and seven runs in the condition SSD, with a flip of a coin in each TS deciding which condition to start with. The task given verbally after the description of the obstacle course read:

“You are one meter from the start line. You are not centered, but start from an unknown position. Your task is to cross the finish line seven meters behind the start line by using the aid. There are eight obstacles on the way which you should not touch. The time required for the run is measured and your contacts with the objects are counted. Contacts caused by the hand controlling the aid are not counted. Time and contacts are equally weighted—do not solely focus on one. You are welcome to think out loud and comment on your decisions, but you won‘t get assistance with finishing the task.”

Contacts with the cane were not included in the statistics, as an essential aspect of its operation is the deliberate induction of contact with obstacles. In addition, for both aids, contacts caused by the hand guiding it were not included in the statistics as well in order to motivate the subjects to freely interact with the aids. There was a clicking sound positioned at the end of the course (centred and 2 m behind the finish line) to roughly guide the direction. There was no help or other type of interference while participants were performing the courses. Only when they accidentally turned more than 90 degrees away from the finish line were they reminded to pay attention to the origin of the clicking sound. Both task completion time and obstacle contacts (including a rating in mild/severe contacts) were entered into a macro-assisted Excel spreadsheet on a hand-held tablet by the experimenter, who was following the subjects at a non-distracting distance. The data of all runs can be found in the study data in Appendix A.

A total of 14 different course layouts were used (Figure 3), seven of which were longitudinal axis mirror images of the other seven. The layout order within one aid condition (SSD/cane) over all TS was the same for all participants and predetermined by drawing all 14 possible variations for each TS without laying back. This means that all participants went through the 14 layouts four times each, but in a different order for each TS and with varying aids, so that a memory effect can be excluded.

The layouts were created in advance using an algorithm that distributed the obstacles over the 50 cm grid. A sequence of 20 of these layouts was then evaluated in self-tests and with pre-subjects, leaving the final seven equally difficult layouts (Figure 3).

The study design and the experimental setup were inspired by a proposal of a standardised obstacle course for assessment of “visual function in ultra low vision and artificial vision” [63] but has been adapted due to spatial constraints and selected study objectives (e.g., testing with two groups and limited task scope only). There are two further studies suggesting a very similar setup for testing sensory substitution devices [64,65] that were not considered for the choice of this study design.

#### 3.5.3. Practice Session

The practice sessions were limited to 15 min and followed a fixed sequence of topics and interaction patterns to be learned with the two aids (Figure 4A–C). In the training sessions obstacles were arranged in varying patterns by the experimenter. Subjects received support as needed from the experimenter and were not only allowed to touch objects in their surroundings, but were even encouraged to do so in order to compare the stimuli perceived by the aid with reality.

In the case of the SSD training, after initially learning the body posture and movement, the main objective was to understand exactly this relationship between stimuli and real object. For this purpose, the subjects went through, for example, increasingly narrow passages with the aim of maintaining a safe distance to the obstacles on the left and right. Later, the tasks increasingly focused on finding strategies to find ways through course layouts similar to the training layouts.

While the training with the white cane (in the sighted group) took place in comparable spatial settings, here the subjects learned exercises from the cane programme within an O&M training (posture, swing of the cane, gait, etc.). The experimenter himself received a basic cane training from an O&M trainer in order to be able to carry it out in the study. The sighted subjects were therefore not trained by an experienced trainer, but all by the same person. At the same time, the SSD was not trained “professionally” either, as there are no standardised training methods specifically for the device yet.

#### 3.5.4. Qualitative Approaches

In addition to the quantitative measurements, the subjects were asked to think aloud, to comment on their actions and to describe why they made certain decisions, both during training and during breaks between trials. These statements were written down by hand by the experimenter.

After completion of the last trial, the subjects were asked to fill out the final questionnaire. It consisted of three parts: firstly, the 10 statements of the System Usability Score (SUS) Test [66] on a 0–4 Likert agreement scale; Secondly, 10 further custom statements on handling and usability on the same 0–4 Likert scale; And finally seven questions on different scales and in free text about perception, suggestions for improvement and the possibility to leave a comment on the study. The questions of part one and two were always asked twice: once for the SSD and once for the cane. The subjects could complete this part of the questionnaire either by handwriting or with the help of an audio survey with haptic keys on a computer. This allowed both sighted and blind subjects to answer the questions without being influenced by the presence of the investigator. The third part, on the other hand, was read out to the blind subjects by the investigator, who noted down the answers.

Due to the small number of participants, the results of the questionnaire are not suitable for drawing statistically significant conclusions, but should rather serve the qualitative comparison of the SSD with the cane and support further developments on this or similar SSDs. In the study data in Appendix A there is a list with all questionnaire items and the Likert scale answers of items 1–20. In the Results section and in Appendix I relevant statements made in the free text questions are included.

### 3.6. Analysis and Statistical Methods

A total of 784 trials in 14 different obstacle course layouts were performed by every subject over all sessions. The dependent variables were *task completion time* (in short *time*) and *number of contacts* (in short *contacts*).

Fixed effects were:*group:* between-subject, binary (blind/sighted)*aid:* within-subject, binary (SSD/cane)*TS:* within-subject, numerical and discrete (the four levels of training)

Variables with random effects were: *layout* as well as the *subjects* themselves, nested within their corresponding group.

The quantitative data of the dependent variable task completion time was analysed by means of parametric statistics using a linear mixed model (LMM). In order to check whether the chosen model corresponds to the established assumptions for parametric tests, the data were analysed according to the recommendation of Zuur et al. [67]. The time variable itself has been normalised in advance using a logarithmic function to meet those assumptions (referred to in the following as *log time*). With the assumptions met, all variables were then tested for their significance to the model and their interactions with each other. See Appendix H for details on the model, its fitting procedure, the assumption and interactions tests and corresponding plots.

Most statistical methods only test for the presence of differences between two treatments and not for their degree of similarity. To test the sub-hypotheses of H1, a non-inferiority test (H1a) and an equivalence test (H1b) were thus carried out. These check whether the least squares (LS) means and corresponding confidence intervals (CI) of a selected contrast exceed a given range (here 25 percentage points in both sub-hypotheses) either in the lower or in the upper direction. In order to confirm the latter, equivalence, both directions must be significant; For non-inferiority only the “worse” (in this case the slower side) has to be significant (since it would not falsify the sub-hypothesis if the SSD were unequally faster) [68].

No statistical tests were performed on the contacts data. Since the data structure is zero-inflated and poisson distributed, non-parametric tests such as a generalised linear mixed model would be required, resulting in low statistical power given the sample size. Nevertheless, descriptive statistical plots of these data alongside the analysis of the log time statistics are to be included in the next section.

All analyses have been executed using the statistical computing environment *R* and the graphical user interface *RStudio*. The *lme4* package was used to run LMMs. To calculate Least LS means, their CI and the non-inferiority/equivalency tests, the *emmeans* and the *emtrends* package was used. In this paper averages are shown as arithmetic mean with the corresponding standard deviation. For all statistical tests an alpha level of 0.05 was chosen.

## 4. Results

### 4.1. Overview

To give an impression of the study procedure, a series of videos was made available in high resolution at https://vimeo.com/channels/unfoldingspace (accessed on 26 January 2022). A selection of lower resolution clips is also attached to Appendix A. The corresponding subject identifier and the trial number can be found in the opening credits and the descriptive text. All test persons shown here have explicitly agreed to the publication of these recordings.

To get an overview of the gathered data, Figure 5 shows log time (the normalised time) for both aid conditions over the four TS horizontally split by group. The plot contains all individual samples as dots in the background, summarised in boxplots including the mean (marked with the “×” sign) in addition to the common median line. The solid line represents the linear regression of the fitted LMM. In this and in following plots the variables aid and group are shown as combination with the levels sighted & SSD (S&S), sighted & cane (S&C), blind & SSD (B&S) and blind & cane (B&C).

Contacts show a similar picture (Figure 6): in the last TS, sighted subjects touched an average of 0.38 ± 0.73 objects per run with the SSD and only 0.12 ± 0.33 with the cane. Blind subjects also showed a comparable response in the last TS, touching an average of 0.45 ± 0.89 objects per run with the SSD, while touching only an average of 0.4 ± 0.63 objects with the cane. As mentioned, these differences cannot be reasonably tested.

### 4.2. Learning Effect

As a basic assumption for the subsequent hypothesis tests, the learning effect on performance has to be investigated. A significant effect of TS on log time was expected in all combinations except B&C, in which the subjects were already familiar with the aid. Still, with habituation to the task and familiarity with the conditions of the course (size, type of obstacles, etc.), a negligible effect could be expected in all four conditions, i.e., also in the case of B&C. The test was carried out by adjusting the base level of the LMM to the four different combinations. TS shows a statistically significant effect on log time in Condition S&S (intercept = 4.37, slope = −0.13, SE = 0.04, *p* = 0.007), S&C (intercept = 4.03, slope = −0.15, SE = 0.04, *p* = 0.002) and in B&S (intercept = 3.91, slope = −0.16, SE = 0.04, *p* = 0.001) but *not* in B&C (intercept = 3.08, slope = −0.06, SE = 0.04, *p* = 0.137). The expected learning progress was thus confirmed by the tests; the general habituation slope over the course of the study for all subjects and aids was around −0.06 s on log scale.

### 4.3. Hypothesis H1|Performance

Given the knowledge of the significant effects of group, aid and TS and their interactions, the two sub-hypothesis H1a and H1b could be tested.

#### 4.3.1. H1a|Non-Inferiority of SSD Performance

In order to accept H1a, two separate tests were carried out: *firstly*, a pairwise comparison using Tukey’s HSD test between conditions S&S and S&C—both under the condition of TS being 4 (after last training): using the Kenward–Roger approximation, a significant difference (*p* < 0.001) was found, with the log time LS means predicted to be 54.43% slower with the SSD (3.87 s, SE = 0.13) than with the cane (3.26 s, SE = 0.13). Back transformed to the response scale, this gives a predicted time of 47.9 s (95% CI [64.2 s, 35.8 s]) for the SSD and 31.0 s (95% CI [41.6 s, 23.2 s]) for the cane to complete one obstacle course run. *Secondly*, the test for non-inferiority between these two conditions (using the Kenward–Roger approximation and the Šidák correction) was performed and found to be non-significant (*p* value 1). This means that the SSD is significantly different from the cane and could be considered inferior within the predefined tolerance range of 25%. H0 of H1a thus *could not be rejected*. The difference between SSD and cane under the condition investigated can also be observed in Figure 7A.

For contacts, as mentioned, a statistical analysis is not feasible. Still, the results can be compared descriptively in previous plot Figure 6. As already mentioned, the difference in measured mean contacts per run differed from 0.38 ± 0.73 objects per run with the SSD and only 0.12 ± 0.33 with the cane.

#### 4.3.2. H1b|Equivalence of Learning Progress

To accept H1b, the learning progress of the SSD had to be compared between the two groups, again by using two tests: *firstly*, the estimated effect of TS on log time differed by only 0.04 s (SE = 0.06) between S&S (−0.12 s, SE = 0.04) and B&S (−0.16 s, SE = 0.04) condition, while not being significant (*p* value = 0.89). This means that there is no proof at this point that the learning progress between the groups is different. *Secondly*, to examine the degree of similarity of the given contrast an equivalence test was carried out (again using Kenward–Roger approximation and Šidák correction): a significant *p* value of 0.016 indicated the presence of equivalence of learning progress in both groups with the SSD (within the predefined tolerance range of 25%). H0 of H1b thus *could be rejected*. The learning progress of the SSD across the groups can also be observed in Figure 7B.

Again for contacts, a statistical analysis is not feasible. Nevertheless, it appears useful to compare the progress of the sighted and blind curve with the SSD in previous plot Figure 6. In particular, the running average described in this figure suggested a quite similar progress between those two.

### 4.4. Hypothesis H2|Usability & Acceptance

In Figure 8, one can find a tabular evaluation and a graphical representation of all Likert scale questions of the first and second part of the questionnaire (including the SUS). In general, one can see that the degree of coverage between SSD and cane was comparatively high. The discussion section therefore looks at the questions that show the greatest average deviations and discusses them in a classifying manner. There is no graphical representation of questions 21–27 as they were in free text or on other scales.

#### 4.4.1. System Usability Score

The System Usability Score, which was queried in the first 10 questionnaire items, results from the addition of all scores multiplied by 2.5 and thus ranges from 0 to a maximum of 100 possible points. The SSD achieved an average SUS of 50 in this study, while the cane scored quite similarly at 53. As expected, the cane performed slightly better in the blind group (54) than in the sighted (52), while the SSD performed better in the sighted (51) than in the blind (49). The differences are rather negligible due to the sample size but can be seen as an indicator of a quite comparable assessment of both systems.

#### 4.4.2. Clustered Topics in Subjects’ Statements

Statements expressed in the free interviews after each session, during the aloud reflection in the training sessions and in the free text questions of the questionnaires were grouped into five main topics (with the number of subjects mentioning them in brackets):**Cognitive Processing of the Stimuli (9).** From the statements it is quite clear that the use of the SSD at the beginning of the study required a considerably higher cognitive effort than the use of the cane (Appendix I, Table A3, ID 1–4). Towards the end of the study, the subjects still reported a noticeable cognitive effort. However, they often also noted that the experience felt different than it did at the beginning of the training and that they could imagine that further training would reduce the effort even more (Appendix I, Table A3, ID 5 & 6). The subjects’ reports towards the end of the study also suggested that deeper and more far-reaching experiences might be possible with the SSD than with the cane (Appendix I, Table A3, ID 7–9).**Perception of Space and Materiality (6).** The topic of how subjects perceived space and its materiality is undoubtedly related to the previously cited statements about the processing of stimuli: it is noteworthy how often spatial or sometimes visual accounts were assigned to the experiences with the SSD, while the cane was rather described as a tool for warning of objects ahead (Appendix I, Table A4).**Wayfinding Processes (5).** It was mentioned as an advantage of the glove that, in contrast to the cane, an obstacle can already be detected before contact is made—i.e., earlier and from a greater distance; A different path can then be taken in advance in order to avoid a collision with this object. In addition, some described an unpleasant feeling of actively bumping into obstacles with the cane just to get information. However, these were mainly sighted people who were not yet used to handling the cane (Appendix I, Table A5).**Enjoyment of Use (3).** The cane is described by some as easy to learn but therefore less challenging and less fun (Appendix I, Table A6).**Feeling Safe and Comfortable (3).** On the other hand, subjects also report that they feel safer and more comfortable with the cane (Appendix I, Table A7).

#### 4.4.3. Advantages of the SSD

In question 25 (Q25) the subjects could name pros and cons of both devices. These were summarised to topics, being described from the perspective of the advantages of the Unfolding Space Glove. An advantage of the cane, for example, was thus evaluated as a disadvantage of the SSD. The most frequently mentioned (by three or four subjects) advantages of the SSD were the following: more spatial awareness is possible; one can survey a wider distance; the handling is more subtle and quiet. Frequent disadvantages were: the higher learning effort for the SSD and the fact that one can obtain less information about the type of objects due to missing acoustic feedback.

#### 4.4.4. Suggestions for Improvement from Subjects

In Q26, the subjects were encouraged to list their suggestions for improvement for a future version of the same SSD—even if these may not be technically feasible. The two biggest wishes addressed two well-known problems of the prototype: detection of objects close to the ground (e.g., steps, thresholds, unevenness, …) was requested by five subjects and the detection of objects closer than 10 cm (where the prototype currently cannot measure and display anything) was requested by four of them. Both would probably have been mentioned even more frequently if they had not already been pointed out as well-known problems at the beginning of the study. Additionally, subjects (number in brackets) wished that they did not have to wear a battery on their arm (3) and wished that the device was generally more comfortable (2). Some individuals mentioned that they would like to customise the configuration (e.g., adjust the range). Some wished for the detection of certain objects (e.g., stairs) or characteristics of the room (brightness/darkness) to be communicated to them via vibration patterns or voice output.

### 4.5. Hypothesis H3|Distal Attribution

H3 could be rejected, as there were no specific indications of distal attribution of perceptions in the subjects’ statements. However, some of the statements strongly suggest that such patterns were already developing in some subjects (Appendix I, Table A8), which is why this topic will be addressed in the discussion.

## 5. Discussion

The results presented above demonstrate not only the perceptibility and processibility of 3D images by means of vibrotactile interfaces for the purpose of navigation, but also the feasibility, learnability and usefulness of the novel Unfolding Space Glove—a haptic spatio-visual sensory substitution system.

Before discussing the results, it has to be made explicitly clear that the study design and the experimental set-up do not yet allow generalisations to be made about real-life navigational tasks for blind people. In order to be able to define the objective of the study precisely, many typical, everyday hazards and problems were deliberately excluded. These include objects close to the ground (thresholds, tripping hazards and steps) or the recognition of approaching staircases. Furthermore, auditory feedback from the cane, which allows conclusions to be drawn about the material and condition of the objects in question, were omitted. In addition, there is the risk of a technical failure or error, the limit of a single battery charge and other smaller everyday drawbacks (waterproofness, robustness, etc.) that the prototype currently still suffers from. Of course, many of the points listed here could be solved technically and could be integrated into the SSD at a later stage. However, they would require development time, would have to be evaluated separately and can therefore not simply be taken for granted in the present state.

With that being said, it is possible to draw a number of conclusions from the data presented. First of all, some technical aspects: the prototype withstood the entire course of the study with no technical problems and was able to meet the requirements placed on it that allow the sensory experience itself to be assessed as independently of the device as possible. These include, for example, intuitive operation of the available functions, sufficient wearing comfort, easy and quick donning and doffing, sufficient battery life and good heat management.

The experimental design can also be pointed out: components such as data collection via tablet, the labelled grid system for placing the obstacles plus corresponding set-up index cards and the interface for real-time monitoring of the prototype enabled the sessions to be carried out smoothly with only one experimenter. An assistant helped to set up and dismantle the room, provided additional support (e.g., by reconfiguring the courses and documented the study in photos and videos), but neither had to be, nor was present at every session. Observations, ratings and participant communication were carried out exclusively by the experimenter.

Turning now to the sensory experience under study, it can be deduced that 3D information of the environment is very direct and easy to learn. Not only were the subjects able to successfully complete all 392 trials with the SSD, but they also showed good results as soon as the first session and thus after only a few minutes of wearing the device. This is, to the best of our knowledge, in contrast to many other SSDs in the literature, which require several hours of training before the new sensory information can be used meaningfully (in return, usually offering higher information density).

Nevertheless, the cane outperforms the SSD in time and contacts, in both groups in the first TS and at every other stage of the study (also see Figure 5 and Figure 6). Apart from the measurements, the fact that the cane seems to be even easier to access than the glove is also shown in the results of the questionnaire among the sighted subjects, for whom both aids were new: while many answers between the SSD and the cane only differed slightly on average (Δ≤ 0.5), the deviations are greatest in questions about the learning progress. Sighted subjects thought the cane was easier to use (Q3, Δ = 1.0), could imagine that “most people would learn to use this system” more quickly (Q7, Δ = 0.8) and stated that they had to learn less things before they could use the system (Q10, Δ = 0.7) compared to the SSD.

### 5.1. Hypothesis 1 and Further Learning Progress

At the end of the study and after about 2 h of wearing time, H1a states that the SSD is still about 54% slower than the cane. Even though this difference in walking speed could be acceptable if (and only if) other factors gave the SSD an advantage, H1 had to be rejected: the deviation exceeds the predefined 25% tolerance range and thus can no longer be understood as an “non-inferior performance”. H1b, however, can be accepted, indicating that due to a “equivalent learning progress” between the groups these results would also apply to blind people who have not yet learned either device.

Left unanswered and holding potential for further research is the question of what the further progression of learning would look like. The fact that the cane (in comparison to the SSD) already reached a certain saturation in the given task spectrum at the end of the study is indicated by several aspects: looking at the performance of B&C, one can roughly estimate the time and average contacts that blind people need to complete the course with their well-trained aid. At the same time, the measurements of S&C are already quite close to those of B&C at the end of the study, so that it can be assumed that only a few more hours of training would be necessary for the sighted to align with them (within the mentioned limited task spectrum of the experimental setup). The assumption that the learning curve of the SSD is less saturated than that of the cane at the end of the study is supported by sighted subjects stating that the cane required less concentration at the end of the study (Q18, 1.8) than the SSD (Q18, 2.7). Furthermore, they expected less learning progress for the cane with “another 3 h of training” (Q23, 2.2) in contrast to the SSD (Q23, 3.5) and also in contrast to the learning progress they already had with the cane during the study (Q22, 3.3). Therefore, a few exciting research questions are whether the learning progress of the SDD would continue in a similar way, at which threshold value it could come to rest and whether this value would eventually be equal to or better than that of the cane. Note that the training time of 2 h in this study is far below that of many other publications on SS; one often-cited study e.g., reached 20–40 h of training with most and 150 h with one particular subject [32].

Another aspect that confounds the interpretation of the data is the presence of a correlation between the two independent variables time/log time and contacts. The reason for this is quite simple: faster walking paces lead to a higher number of contacts. A slow walking pace, on the other hand, allows more time for the interpretation of information and, when approaching an obstacle, to come to a halt in time or to correct the path and not collide with the obstacle. The subjects were asked to consider time and contacts as being equivalent. Yet these variables lack any inherent value that would allow comparing a potential contact with loss of time, for example. Personal preference may also play a role in the weighting of the two variables: fear of collisions with objects (possibly overrepresented in sighted people due to unfamiliarity with the task) may lead to slower speeds. At the same time, the motivation to complete the task particularly quickly may lead to faster speeds but higher collision rates. Several subjects reported that towards the end of the study, they felt that they had learned the device to the point where contacts were completely avoidable for them given some focus. This attitude may have led to a bias in the data, which can be observed in the fact that time increased in the sighted group with both aids towards the end of the study while the collisions continued to fall. It seems to be difficult to solve this problem only by changing the formulation of the task and without expressing the concrete value of an obstacle contact in relation to time (e.g., a collision would add 10 s to the time or leads to the exclusion of this trial from the evaluation).

### 5.2. Usability & Acceptance

While there is an ongoing debate about how to interpret SUS and what value the scoring system has in the first place, the scores of the SSD (50) and the cane (53) are comparably low. Therefore they can be interpreted as being “OK” only and, in the context of all the systems tested with this score, they tend to be in the 20% or even 10% percentile [69]. It should be noted, however, that the score is rarely used to evaluate assistive devices at all, which may partly explain a generally poorer performance in those. The score is, however, suitable to “compare two versions of an application” [69]: the presented results therefore indicate that usability in the two tested aids does not fundamentally differ in the somewhat small experimental group. This equivalence can also be assessed from other questionnaire items with most having very few deviations:

Looking at the Likert scale averages of the entire study population (blind & sighted), biggest deviations (Δ ≥ 0.5) can be observed in the expected recognisability of a “visual impairment because of the aid” that is stated to be much lower (Q15, Δ = 2.0) with the SSD (Q15, 1.8) than with the cane (Q15, 3.8). Just as in the sighted group, the average of both groups stated that the cane was easier to use (Q3, Δ = 0.8) and required less concentration at the beginning of the study (Q17, Δ = 0.8), whereas this difference becomes negligible by the end of the study (Q18, Δ = 0.3). Last but not least, the two aids differed in Q11 (“I had a lot of fun using the aid”), in which the SSD scored Δ = 0.7 points better.

Exemplary statements have already been presented in the results section, summarised and classified into topics. They support the theses that the Unfolding Space Glove achieves its goal of being easier and quicker to learn than many other SSDs while providing users with a positive user experience. However, the sample size and the survey methods are not sufficient for more in-depth analyses. The presentation should rather serve the purpose of completeness and provide insights into how the learning process was perceived by the test persons in the course of the study.

### 5.3. Distal Attribution and Cognitive Processing

The phenomenon of distal attribution (sometimes externalisation of the stimulus) in simplified terms describes when users of an SSD report to no longer consciously perceive the stimulus at the application site on the body (here e.g., the hand), but instead refer to the perceived objects in space (distal/outside the body). This can also be observed, for example, when sighted people describe visual stimuli and do not describe the perception of stimuli on their retina, but instead the things they *see* at their position in space. Distal attribution was first mentioned in early publications of Paul Bach-y-Rita [31,33] in which participants received 20–40 h of training (one individual even 150 h) with a TVSS and has also been described in other publications e.g., about AVSS devices [40]. Ever since it has been discussed in multifaceted and sometimes even philosophical discourses and has been topic of many experimental investigations [70,71,72,73]. As already described in the results section, the statements do not indicate the existence of this specific attribution. However, they do show a high degree of spatio-motor coupling of the stimuli and suggest the emergence of distal-like localisation patterns. The wearing time of only 2 h, however, was comparatively short and studies with longer wearing times would be of great interest on this topic.

### 5.4. Compliance with the Criteria Set

In the introduction to this paper, 14 criteria were defined that are important for a successful development of an SSD. See also Appendix B for a description of those. Chebat et al. [55], who collected most of them, originally did so to show problems of known SSD proposals. In the design and development process of the Unfolding Space Glove these criteria did play a crucial role from the very start. Now, with the findings of this study in mind, it is time to examine to what degree it can meet the list of criteria by classifying six of its key aspects (with numbers of the respective criteria in parentheses):**Open Source and Open Access.** The research, the material, the code and all blueprints of the device, are open source and open access. In the long run, this can lead to lower costs (1) of the SSD and a higher dissemination (4), as development expenses are already eliminated; the device can theoretically even be reproduced by the users themselves.**Low Complexity of Information.** The complexity and thus the resolution of the information provided by an SSD is on a low level in order to offer low entry barriers when *learning* (1) the SSD. The *user experience and enjoyment* (13) is not affected much by the *cognitive load* (5). This requirement is of course in contrast to the problems of low *resolution* (9).**Usage of 3D Input.** The use of *spatial depth* (7) images from a 3D camera inherently provides suitable information for locomotion, reduces the *cognitive load* (5) of extracting them from conventional two-dimensional images and is independent of lighting situations and bad *contrast* (8).**Using Responsive Vibratory Actuators as Output.** Vibration patterns can be felt (in different qualities) all over the body, are non-invasive, non-critical and have no medically proven harmful effect. Linear resonance actuators (LRA), offer low *latency* (3), are durable, do not heat up too much and are still *cost* (10) effective, although they require special integrated circuits to drive them.**Positioning at the Back of the Hand.** The site of stimulation on the back of the hand, although disadvantaged by other factors such as total available surface or density of vibrotactile receptors [74], proved to be a suitable site of stimulation with regard to several aspects: a fairly natural posture of the hand when using the device enables a discrete body posture, does not interfere with the overall *aesthetical appearance* (14) and *preserves sensory and motor habits* (12). The *orientation of the Sensor* (6) on the back of the hand is hoped to be quite accurate as we can use our hands for detailed motor actions and have a high proprioceptive precision in our elbow and shoulder [75]. Last but not least, the hand has a *high motor potential* (11) (rotation and movement in three axes), facilitating the sensorimotor coupling process.**Thorough Product & Interaction Design.** A good design does not only consist of the visible shell. Functionality, interaction design and product design must be considered holistically and profoundly, and in the end they pay off on many aspects apart from the *aesthetic appearance* (14) itself, such as almost all of the key aspects discussed in this section and on *user experience and joy of use* (13).

## 6. Conclusions

The Unfolding Space Glove, a novel wearable haptic spatio-visual sensory substitution system, has been presented in this paper. The glove transforms three-dimensional depth images from a time of flight camera into vibrotactile stimuli on the back of the hand. Blind users can thus haptically explore the depth of the space surrounding them and obstacles contained therein by moving their hand. The device, in its somewhat limited functional scope, can already be used and tested without professional support and without the need of external hardware or specific premises. It already is highly portable and offers a continuous and very immediate feedback, while its design is unobtrusive and discreet.

In a study with eight blind and six sighted (but blindfolded) subjects, the device was tested and evaluated in obstacle courses. It could be shown that all subjects were able to learn the device and successfully complete the parcours presented to them. Handling has low entry barriers and can be learned almost intuitively in a few minutes, with the learning progress between blind and sighted subjects being fairly comparable. However, at the end of the study and after about 2 h of wearing the device, the sighted subjects were significantly slower (by about 54%) in solving the courses with the glove compared to the white long cane they had worn and trained for the same amount of time.

The device meets many basic requirements that a novel SSD has to fulfil in order to be accepted by the target group. This is also reflected in the fact that the participants reported a level of user satisfaction and usability that is—despite its different functions and complexity—quite comparable to that of the white long cane.

The results in the proposed experimental set-up are promising and confirm that depth information presented to the tactile system can be cognitively processed and used to strategically solve navigation tasks. It remains open how much improvement could be achieved in another two or more hours of training with the Unfolding Space Glove. On the other hand, the results are of limited applicability to real-world navigation for blind people: too many basic requirements for a navigation aid system (e.g., detection of ground level objects) are not yet included in the functional spectrum of the device and would have to be implemented and tested in further research.

## Figures and Tables

**Figure 1 sensors-22-01859-f001:**
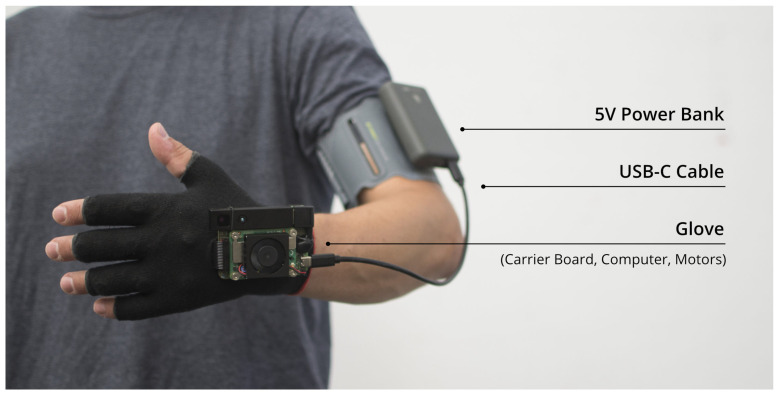
The final prototype with labeled components.

**Figure 2 sensors-22-01859-f002:**
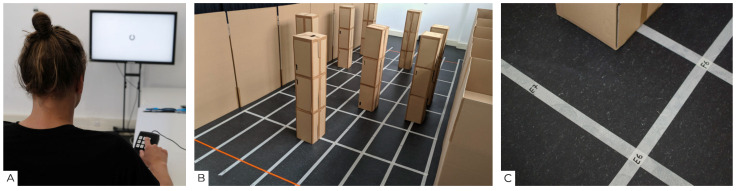
(**A**) Carrying out the “FrACT” Landolt visual acuity test with screen and tactile input device (lower right corner). (**B**) The obstacle course in the study room. (**C**) Close up of the grid system used for quick rebuilding of the layouts.

**Figure 3 sensors-22-01859-f003:**
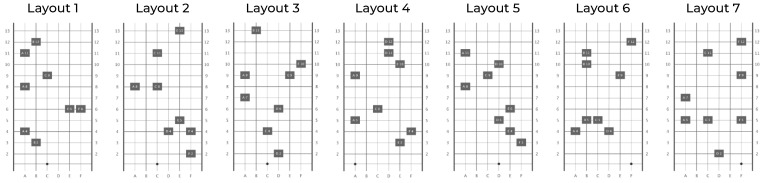
The seven obstacle course layouts used in the study (in non-mirrored variant).

**Figure 4 sensors-22-01859-f004:**
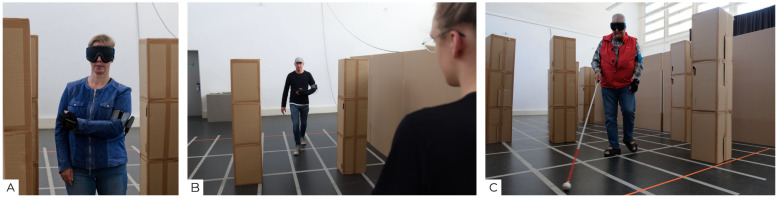
(**A**) A subject practising with the SSD in the tutorial at the beginning of the study (**B**) A typical task in the practice sessions: passing a gap between two obstacles (**C**) A sighted subject training with the cane during a practice session.

**Figure 5 sensors-22-01859-f005:**
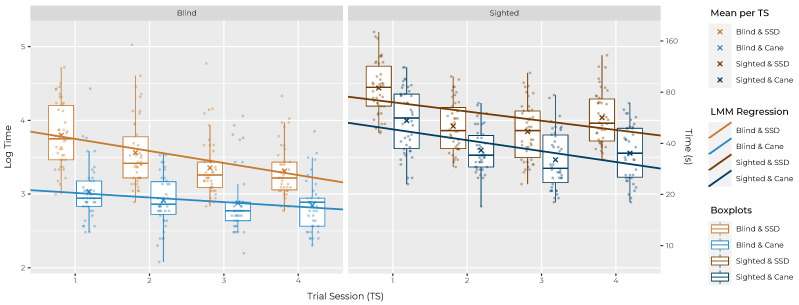
Trial session (TS) vs. normalised task completion time (log time) of all samples as boxplots, means and regression lines from the linear mixed model. Split horizontally by group (blind/sighted) and subdivided by aid in colour (SSD/cane).

**Figure 6 sensors-22-01859-f006:**
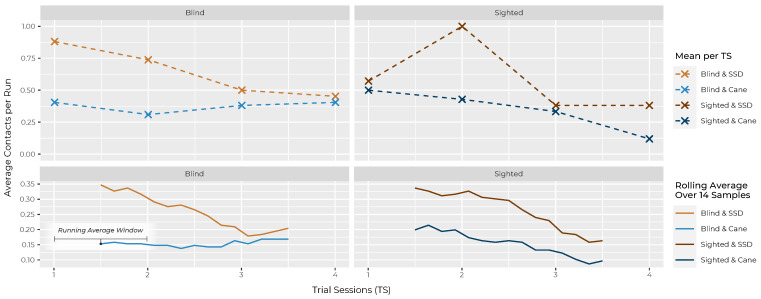
Trial sessions (TS) vs. average number of contacts with obstacles per run. The upper half shows the means per TS; The lower one shows a running average (running over all samples) calculated to better visualise trends. Note the wide window of 14 samples per data point. The plot is split horizontally by group (blind/sighted) and subdivided by aid in colour (SSD/cane).

**Figure 7 sensors-22-01859-f007:**
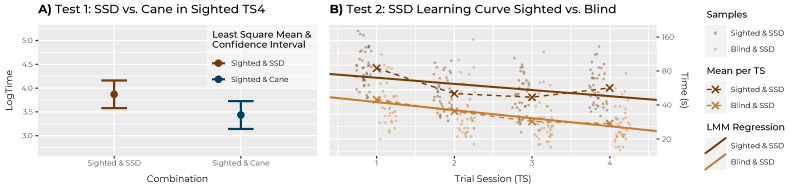
Graphical representations of the two main hypothesis tests. (**A**) Sub-hypothesis H1a: performance with the two devices (SSD/cane) after the last training (only Trial Session 4). Least square means and the corresponding confidence interval extracted from the linear mixed model are shown to demonstrate the test. (**B**) Sub-hypothesis H1b: the learning curves with the SSD in both groups (sighted/blind) over the course of the training (all Trial Sessions) are shown. The test only compares the linear regression slopes, but means and the underlying data points are plotted here as well.

**Figure 8 sensors-22-01859-f008:**
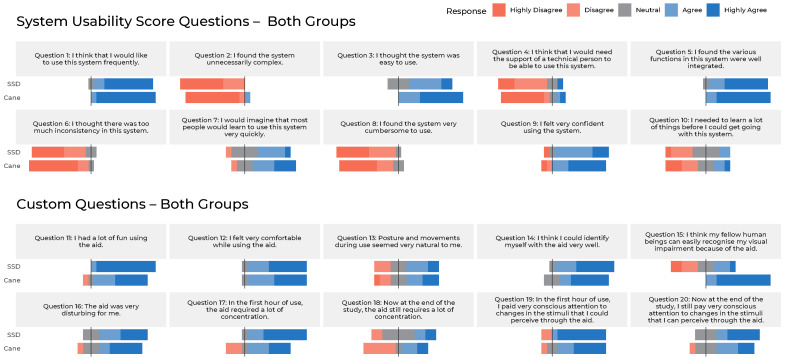
A summary of all Likert scale questionnaire items. The results are summarised for both groups (sighted/blind) and split per question according to SSD and cane.

## Data Availability

The data presented in this study are available in Appendix A at: https://www.mdpi.com/article/10.3390/s22051859/s1.

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
