# Peer review of "The Unfolding Space Glove: A Wearable Spatio-Visual to Haptic Sensory Substitution Device for Blind People"

_sensors, 2022, doi:10.3390/s22051859_

Round 1
Reviewer 1 Report
In this study, the authors tested the training effect of a wearable device, the unfolding space glove, for assisting the spatial navigation of the blind people. The results did show the training effect, but the achieved performance by the glove was comparable to the performance by the traditional white cane. Both the experimental design and data analysis are appropriate, and the manuscript is well written. The findings will inspire future empirical studies as well as device improvement. I have only a few minor issues before considering the publication.
1) It would be nice if the demographical information (e.g., age, the experience of using white cane) of the two groups of participants can be presented in a table. Given that the sample size is small, some of the results might be due to the individual differences.
2) Some statements concerning the results are not presented with statistical evidence. For instance, Line 473-477, no statistical power of the claimed difference was provided.
Author Response
In Response to Reviewer 1
Dear Reviewer,
thank you very much for taking the time to review and comment upon our paper manuscript on the Unfolding Space Glove. We are very pleased about your general approval of the paper and would like to respond to your - in our opinion - well-founded and constructive feedback in the following.
With best thanks and sincere regards, Jakob Kilian (corresponding author)
Point 1: It would be nice if the demographical information (e.g., age, the experience of using white cane) of the two groups of participants can be presented in a table. Given that the sample size is small, some of the results might be due to the individual differences.
Response: We appreciate this criticism! We have included a summarising table with data on the subjects in the appendix (F) and linked it in the main text, whereas previously this information was only available in the supplementary files.
Furthermore, we believe that individual differences do not confound the results; on the contrary, we believe that the variance in the groups improves the statistical model. Our study was designed such that the direct performance comparison (H1a) was only carried out in group B (sighted) and thus (with regard to cane use) with naïve subjects who were newly learning both aids. The blind group then ( H1b) serves to compare the learning curve of the SSD between both groups and - of course - to collect qualitative feedback from those who would eventually use such a device. Then, a linear mixed model was used for the evaluation, which allows testing for fixed effects between certain variables, while still taking into account random effects of other variables (e.g. the subject’s basic navigation performance). In this way, the varying basic navigation skills of the subjects are already taken into account and generally slower or poorer performers would not pose a problem for the scenario to be tested.
We hope that we have been able to clarify this issue with this brief explanation and that the additional table sheds light on the composition of the study population.
Change: ll. 281-282 & ll. 1050-1054 | Comment A1
Point 1: Some statements concerning the results are not presented with statistical evidence. For instance, Line 473-477, no statistical power of the claimed difference was provided.
Response: Thank you for pointing this out. We agree with you that the values presented here are confusing because they are not based on a statistical model. It was rather intended to give a general overview of the average times needed per run in the different combinations. However, since only values of the statistical model are presented in the further course of the paper and the informative value of this overview is justifiably doubtful, we have removed this paragraph.
We could not find any other presented values that were not statistically supported and would be very happy to receive a report of more of these incidents.
Change: l. 480 | Comment A2

Reviewer 2 Report
Overview and Scientific Merit.
This paper describes the design, development, and initial evaluation of a visual to haptic SSD glove that is intended to support blind and visually impaired (BVI) people to experience depth in surrounding 3D space. Some key benefits of the prototype are that it is designed to be portable, real time, robust in usage scenarios, and unobtrusive to the user. These are important design factors that are often ignored in assistive technology development and the authors are lauded for these important design considerations from the onset. Evaluations were conducted both with BVI Ss (8, but really 6 as 2 were not analyzed) and blindfolded-sighted Ss (6), which is appropriate in terms of number for usability evaluations at this stage of development. Importantly, the glove was successful in allowing all participants to complete the trials, although performance was not better than the traditional long cane (which is not that surprising, as I describe more below). Importantly, user input was positive and helped guide future design (although I’d like this described a bit more).
Although there have been many SSD devices developed over the years, most do not succeed as they either fall into what I call the engineering trap (being designed based on engineering requirements and uninformed designers, rather than knowledge of the end user and their actual problems), or because they are providing a solution to a non-existent problem. In some ways, this device is meant to solve an already solved problem—that of the long cane. However, it does provide some important augmentations to the traditional cane, and the authors have made a good effort to consider important design decisions and to include BVI feedback, which is important. They also clearly understand many of the motivating factors to SSDs, which is also critical (although I think they could better discuss BVI navigation and O&M, as discussed below).
I am supportive of the paper but do have a number of things that I would want clarified before final publication, including more explanatory power around the background issues. These are described in detail below and if carefully addressed, I think this would lead to an informative paper that would be of interest to the journal’s readership.
Points to Consider in a revision.
- In the abstract, it would be helpful to describe a bit more the purpose of the glove and in what situations it would be used. What information is being provided to support what types of tasks?
- I am not sure that the best metric of success is whether the glove is better than the cane. First, the cane is a tried and true mobility aid and most BVI people do not want to replace it—they want to augment it, which would be a stronger pitch for this product. Indeed, most assistive mobility devices fail—and there have been hundreds since the 1960s, as the developers want to design something that replaces the long cane or dog guide, rather than to complement/augment it. If acceptance of the device is desired by the target demographic, this long history is worth noting and considering. Second, given the difference in practice—years for many people with the cane and only a short study with the glove, it is neither surprising nor abundantly meaningful that performance was better with the cane. As such, I find your key hypothesis to be poorly motivated am thus, am not surprised that you did not reject the null. This does not mean that the glove is a failure and you would be better able to emphasize this if you started with more realistic assumptions.
- In the intro, I would give a bit more discussion of orientation and mobility (O&M) as relates to BVI navigation. The generic definitions of navigation that you give are fine, but they do not tie to the BVI literature or describe the specific challenges experienced when navigating without vision. Your focus here is on mobility but this should be further discussed. A couple good background references on this issue, and blind spatial cognition more generally are:
Giudice, N. A. (2018). Navigating without Vision: Principles of Blind Spatial Cognition. In D. R. Montello (Ed.), Handbook of behavioral and cognitive geography (pp. 260-288). Cheltenham, UK; Northampton, MA: Edward Elgar Publishing.
Schinazi, V. R., Thrash, T., & Chebat, D. R. (2016). Spatial navigation by congenitally blind individuals. Wiley Interdisciplinary Reviews: Cognitive Science, 7(1), 37-58.
- You argue that guide dogs are expensive, which is a limitation. There are challenges with dogs as mobility aids, which others have discussed, but the expense is not generally one of them for the BVI end-user. Although schools often say the cost of breeding, training, and placement of a dog is north of $40K, this is all from fund raising-the schools are all nonprofits, and the cost to the BVI end-user is nothing or negligible (on the order of $100-$200) depending on the school. The maintenance costs can add up but again, some schools also cover this.
- The authors are congratulated for starting their SSD design based on prior suggestions and pre-requisites for SSD design—this is good procedure and helps to avoid the engineering trap.
- Although using visual etiology and extent of vision loss is a fine (and traditional) way to characterize the blind Ss sample, it says little about the BVI group’s ability and experience with navigation and mobility, which is more relevant here than anything else. It would be more meaningful to clarify in text their extent of O&M training, amount that they independently travel, if they are cane or dog users, etc. If available, the authors are urged to list these characteristics. My guess is that the 2 Ss that were dropped, a troubling number given that this was 25% of the total original sample, is that these people had poor mobility skills/training and did not reliably travel on their own. Conversely, I wager the best had high values on these factors. From my experience running studies for more than 20 years with this demographic, these factors are far more meaningful indicators of success or not with navigation, ETAs, and the like than any medical model classifications of vision loss. This issue is also discussed more in the above 2 references I gave under point 3.
- Maybe I am misunderstanding but… I think the participants were told to not contact any of the obstructions/obstacles along the route and that this occurrence was counted as an error. If so, this is confusing and does not make sense for tradition use of the white cane, which is meant to be swept and to contact obstacles in the process of good cane operation. This contact, and the associated echoes, is critical for identifying the presence, distance, and direction of said obstructions. Thus, did these normal contact instances bias the glove in your analysis? Please clarify.
- The paper is quite long and might benefit from some reduction and condensing. I like the detail provided in the methods (which could use a bit more detail as discussed above) and the results. However, I think you could likely trim and condense a bit in the intro sections to shorten the overall ms.
- This paper was very well written, researched, and covered a lot of relevant topics underlying the current device. This is often not the case with newly designed products and the authors are commended for their work and interesting research.
Author Response
In Response to Reviewer 2
Dear Reviewer,
thank you very much for taking the time to review and comment upon our paper manuscript on the Unfolding Space Glove. We consider the advice to be very elaborate and constructive and would like to address all points individually in the following section.
With best thanks and sincere regards, Jakob Kilian (corresponding author)
Point 1: In the abstract, it would be helpful to describe a bit more the purpose of the glove and in what situations it would be used. What information is being provided to support what types of tasks?
Response: We have rewritten the abstract to convey the purpose and task more clearly right at the beginning. Thank you for this suggestion!
Change: ll. 1-15 | Comment B1
Point 2: I am not sure that the best metric of success is whether the glove is better than the cane. First, the cane is a tried and true mobility aid and most BVI people do not want to replace it—they want to augment it, which would be a stronger pitch for this product. Indeed, most assistive mobility devices fail—and there have been hundreds since the 1960s, as the developers want to design something that replaces the long cane or dog guide, rather than to complement/augment it. If acceptance of the device is desired by the target demographic, this long history is worth noting and considering. Second, given the difference in practice—years for many people with the cane and only a short study with the glove, it is neither surprising nor abundantly meaningful that performance was better with the cane. As such, I find your key hypothesis to be poorly motivated am thus, am not surprised that you did not reject the null. This does not mean that the glove is a failure and you would be better able to emphasize this if you started with more realistic assumptions.
Response: We are aware that the prototype in its current form cannot replace the cane in any way, not least because of missing functionality such as the described lack of detection of objects close to the ground. We decided to test both SSD and cane in the study in order to be able to compare the two forms of information dissemination and to discuss them in respect to the selected controlled range of tasks. The glove in fact was designed to support and augment a cane held in the other hand. In the present study, however, we wanted an isolated assessment of the glove's ability to support navigation tasks and therefore decided to leave out the combination of both aids due to unknown interactions. Apparently we were not able to clarify our intention behind the study design to the fullest extent, which is why we have implemented the changes below. Thank you for pointing this out.
Change: ll. 244-251 | Comment B2
Point 3: In the intro, I would give a bit more discussion of orientation and mobility (O&M) as relates to BVI navigation. The generic definitions of navigation that you give are fine, but they do not tie to the BVI literature or describe the specific challenges experienced when navigating without vision. Your focus here is on mobility but this should be further discussed. A couple good background references on this issue, and blind spatial cognition more generally are:
Giudice, N. A. (2018). Navigating without Vision: Principles of Blind Spatial Cognition. In D. R. Montello (Ed.), Handbook of behavioral and cognitive geography (pp. 260-288). Cheltenham, UK; Northampton, MA: Edward Elgar Publishing. Schinazi, V. R., Thrash, T., & Chebat, D. R. (2016). Spatial navigation by congenitally blind individuals. Wiley Interdisciplinary Reviews: Cognitive Science, 7(1), 37-58.
Response: Thank you for pointing out the very interesting references on the topic of navigation for blind and visually impaired people. We were pleased to include these in the introduction and believe that they support the manuscript, as the prototype presented here is indeed not intended to simply replace vision, but to provide a slice of the spatial environment that subjects can anticipate, learn and use for navigation tasks using their own strategies and approaches.
Change: ll. 52-65 | Comment B3
Point 4: You argue that guide dogs are expensive, which is a limitation. There are challenges with dogs as mobility aids, which others have discussed, but the expense is not generally one of them for the BVI end-user. Although schools often say the cost of breeding, training, and placement of a dog is north of $40K, this is all from fund raising- the schools are all nonprofits, and the cost to the BVI end-user is nothing or negligible (on the order of $100-$200) depending on the school. The maintenance costs can add up but again, some schools also cover this.
Response: Thanks for pointing this out. For the sake of length, we removed the reference to the costs for the end user and instead refer in general to the discussion of pros and cons of the guide dog. A deeper discussion of them would, however, go beyond the scope of this introductory section.
Change: ll. 79-81 | Comment B4
Point 5: The authors are congratulated for starting their SSD design based on prior suggestions and pre-requisites for SSD design—this is good procedure and helps to avoid the engineering trap.
Response: We are particularly pleased about this commendation.
Point 6: Although using visual etiology and extent of vision loss is a fine (and traditional) way to characterize the blind Ss sample, it says little about the BVI group’s ability and experience with navigation and mobility, which is more relevant here than anything else. It would be more meaningful to clarify in text their extent of O&M training, amount that they independently travel, if they are cane or dog users, etc. If available, the authors are urged to list these characteristics. My guess is that the 2 Ss that were dropped, a troubling number given that this was 25% of the total original sample, is that these people had poor mobility skills/training and did not reliably travel on their own. Conversely, I wager the best had high values on these factors. From my experience running studies for more than 20 years with this demographic, these factors are far more meaningful indicators of success or not with navigation, ETAs, and the like than any medical model classifications of vision loss. This issue is also discussed more in the above 2 references I gave under point 3.
Response:
We appreciate this criticism! We have included a summarising table with data on the subjects in the appendix (F) and linked it in the main text, whereas previously this information was only available in the supplementary files. Nevertheless, we cannot provide further information on, for example, the extent of O&M trainings or figures on independently travel habits, as these have not been collected. It also remains questionable how these data could be gathered and to what extent they would be comparable, since the subjects have very different biographies (e.g. when they learned to use the cane) and the topic of mobility would certainly be strongly influenced by other factors such as age.
At this point, we can only emphasise that both individuals have had cane experience for many years and are independently mobile. The fact that their basic navigation performance is worse than that of other participants can even be determined on the basis of the measurement data, insofar as this makes sense due to the limited data available. However, due to the chosen study design, this alone would not be a reason to exclude them from the evaluation: The direct performance comparison (H1a) was only carried out in group B (sighted) and thus (with regard to cane use) with naïve subjects who were newly learning both aids. The blind group then ( H1b) serves to compare the learning curve of the SSD between both groups and - of course - to collect qualitative feedback from those who would eventually use such a device. Furthermore, a linear mixed model was used for the evaluation, which – in simple terms – allows testing for fixed effects between certain variables, while still taking into account random effects of other variables. In this way, the varying basic navigation skills of the subjects are already taken into account and generally slower or poorer performers would not pose a problem for the scenario to be tested. Rather, the two subjects were excluded due to the behaviour described and the failure to complete the tasks as expected, so that the statistical relevance of the model would be endangered if they were included in the data set.
We hope that we have been able to clarify this issue with this brief explanation and that the additional table sheds light on the composition of the study population.
Change: ll. 281-282 & ll. 1050-1054 | Comment B6
Point 7: Maybe I am misunderstanding but... I think the participants were told to not contact any of the obstructions/obstacles along the route and that this occurrence was counted as an error. If so, this is confusing and does not make sense for tradition use of the white cane, which is meant to be swept and to contact obstacles in the process of good cane operation. This contact, and the associated echoes, is critical for identifying the presence, distance, and direction of said obstructions. Thus, did these normal contact instances bias the glove in your analysis? Please clarify.
Response: In fact, this seems to be a misunderstanding, which is why we have added the sentence below to the manuscript. Until now, this was only mentioned in the task description in chapter 3.5.2. Contacts of the cane itself were not counted. In addition, those contacts that were made with the hand that operates the aid were not counted as well. Thus, no contacts that were triggered by the execution of the oscillating movement with the cane or the SSD were included in the statistics.
Change: ll. 356-359 | Comment B7
Point 8: The paper is quite long and might benefit from some reduction and condensing. I like the detail provided in the methods (which could use a bit more detail as discussed above) and the results. However, I think you could likely trim and condense a bit in the intro sections to shorten the overall ms.
Response: Thank you for this comment as well - we absolutely agree with you. We have shortened several paragraphs in the introduction in order to reduce the overall length of the manuscript.
Change: multiple locations
Point 9: This paper was very well written, researched, and covered a lot of relevant topics underlying the current device. This is often not the case with newly designed products and the authors are commended for their work and interesting research.
Response: We are delighted about this comment, thank you very much!

Round 2
Reviewer 2 Report
I reviewed an earlier version of this ms and am quite satisfied with the revisions and modifications made to this version of the paper. The authors addressed all of my concerns and revised the ms accordingly, with the review process leading to a much improved and more comprehensive final product. Specifically, clarification of some previously confusing points is well done, addition of citations and connection of current work to other solutions is improved, and greater explanatory information about underlying theory, rationale, and methodology make the paper much stronger. I think the research and write-up is timely, interesting, and insightful and I am happy for it to be published in its current for.